# Green-Based Antimicrobial Hydrogels Prepared from Bagasse Cellulose as 3D-Scaffolds for Wound Dressing

**DOI:** 10.3390/polym11111846

**Published:** 2019-11-08

**Authors:** Yuanfeng Pan, Xiao Zhao, Xiaoning Li, Pingxiong Cai

**Affiliations:** 1Guangxi Key Lab of Petrochemical Resource Processing and Process Intensification Tech., School of Chemistry and Chemical Eng., Guangxi University, Nanning 530004, China; 18376747727@163.com; 2Department of Environmental Science and Engineering, North China Electric Power University, Baoding 071003, China; 1182102064@ncepu.edu.cn; 3College of Petroleum and Chemical Engineering, Beibu Gulf University, Qinzhou 535006, China; pxcai7080@sina.com

**Keywords:** cellulose, guanidine-based polymer, hydrogel dressing, antimicrobial activity

## Abstract

Developing the ideal biomaterials for wound dressing still remains challenging nowadays due to the non-biodegradable features and the lack of antimicrobial activity of conventional synthetic polymer-based dressing materials. To tackle those problems, a novel and green-based antimicrobial hydrogel dressing was synthesized in this work via modifying sugarcane bagasse cellulose with guanidine-based polymer, followed by crosslinking antimicrobial-modified cellulose with unmodified one at various ratios. The resulting hydrogels were comprehensively characterized with swelling measurements, compression test, Fourier transform infrared spectroscopy, and scanning electron microscopy. The results indicated that the dressing possessed the degree of swelling up to 2000% and the compress strength as high as 31.39 Kpa, at 8:2 ratio of pristine cellulose to modified cellulose. The antibacterial activities of the dressing against *E. coli* were assessed using both shaking flask and ring diffusion methods. The results demonstrated that the dressings were highly effective in deactivating bacterium without leaching effect. Moreover, these hydrogels are biocompatible with live cell viability responses of (NIH3T3) cells above 76% and are very promising as wound dressing.

## 1. Introduction

As the natural barrier on the surface of human body, skin plays a vital role in participating in metabolic processes, in regulating body temperature, and in protecting various tissues and organs in the body from being attacked by physical, mechanical, chemical actions, and pathogenic microorganisms [1,2]. Once the skin is injured or damaged by the burn or other factors such as chronic wounds diseases and trauma, the umbrella of our body disappears. Consequently, the physiological functions of the internal organs are deteriorated, and the organs themselves are prone to microbial infection. The multidrug-resistant bacterial infections could further aggravate the wound, leading to serious problems in the clinic [3]. Traditional dressing, such as fine belt, cotton, and gauze are the most widely used types of dressing in clinical practice at present. The main function is to keep the wound dry and prevent harmful bacteria from entering the wound to cause infection by absorbing and evaporating the wound secretion. However, the adhesion of such a dressing to the wound is often too strong, thus causing the secondary injury when dressing is removed or changed. Moreover, its ability of absorbing seepage is limited, so that the protective function of barrier is often lost after being soaked, which causes exogenous infection. Therefore, the innovative wound dressing with improved performance is urgently needed. It is well known that the medical dressing not only protects the wound, but also benefits the recovery of dermal and epidermal tissues for healing. Hence, the ideal dressing should possess the following characteristics: absorbable wound exudate, maintaining the temperature and humidity of the wound surface, good permeability, high antibacterial, anti-inflammatory activities, etc. [4,5].

Among various potential materials suiting the requirements of wound dressing, hydrogel is the one close to an ideal dressing. As a hydrophilic macromolecular compound, hydrogel can hold a large amount of water without dissolution and provide a moist environment for the wound. Crosslinked by covalent bonds, ionic bonds, or hydrogen bonds, hydrogel is rendered a porous and three-dimensional network structure, which enhances its capability holding water significantly [6,7]. Its porous structure can also provide transport channels for low molecular weight solutes, nutrients, and cell waste, which are critical for cell proliferation [8,9,10,11]. The ideal cell carrier material should be able to simulate the natural environment around cartilage tissue matrix, possess certain compressive strength, and have good compatibility with biological tissue. Therefore, the research related to biomedical applications, such as drug carriers for controlled release, tissue scaffolds, and wound dressing has received enormous attention recently [12,13,14]. Hydrogels that are based on natural polymers have many characteristics of extracellular matrix, which can direct cell migration, growth, and tissue during tissue regeneration [9,15,16].

Natural polymers include various polysaccharides, proteoglycans and proteins; and typical examples are alginates, chitosan, cellulose, lignin, heparin, chondroitin collagen, gelatin, fibrin, keratin etc. [17,18] Those materials have been broadly used in wound treatment because of their excellent biocompatibility, biodegradability and similarity to macromolecules that are usually recognized by the human body [16,19,20,21]. As one of the natural biocompatible polysaccharides, cellulose is a sustainable, nontoxic and most abundant natural polymer with good biocompatibility, hydrophilicity, relatively high thermo-stability, high sorption capacity, and cost-effectiveness [22,23]. These properties make cellulose an excellent natural source for the production of hydrogels and composites for various biomedical applications, including wound dressing, drug delivery, and tissue engineering [24,25,26].

Although the hydrogels synthesized from cellulose have many advantages, hydrogels themselves lack antibacterial activity. The ability of functional dressings to resist the infection of harmful microorganisms is particularly important. However, the deactivation caused by most existing antibiotics is based on their impact on the metabolism of bacteria, which already raised concerns related to drug resistance [27]. Killing bacteria based on physical interactions, which would not cause bacterial drug resistance, is the approach to tackle the concerns. It has been proved that the antimicrobial agents with positively charged groups such as quaternary ammonium, or phosphonium, quaternary biguanidine antibiotics, deactivate bacteria via physical interaction [28,29,30]. However, when the antibacterial agent is simply mixed or impregnated into the dressing, the leaching-out of antimicrobial agent can be harmful or can lower the effectiveness. 

To overcome the aforementioned problems, cellulose is chemically modified by incorporating functional groups, to render cellulose antibacterial. The graft copolymerization method has been often adopted in modifying cellulose and investigated extensively in the past decades [23,31]. As a kind of cationic polyelectrolytes, guanidine-based polymer can adsorb on the negative charged surface of bacterial cells by electrostatic interaction and exchange ions with K^+^ and Mg^2+^ in the cell membrane, which in turn changes the fluidity and permeability of the cell membrane, thus destroying the cell membrane and deactivating bacteria. Guanidine salt oligomer is water-soluble and has high efficiency and broad-spectrum inhibition, high killing ability of microorganism, and low mammalian toxicity. However, the research and application of guanidine salt oligomer or polymer in dressing field has been seldom reported [32].

In this work, we aimed at synthetizing antimicrobial hydrogel by grafting guanidine-based polymers onto cellulose fibers via in situ-grafting, using epichlorohydrin as a coupling agent. The covalent bonding induced by crosslinking eliminates the problems associated with leaching and migration of antimicrobial polymer, which is often encountered in the systems by simply adding the antibacterial agent to the dressing via blending or impregnating. The results demonstrated that the dressing consisting of antimicrobial-modified cellulose hydrogel possesses high swelling, excellent antibacterial activity, and is promising for various medical treatments such as wound healing and the repair of damaged tissue.

## 2. Materials and Methods

### 2.1. Materials

1,6-hexanediamine and guanidine hydrochloride were obtained from Aladdin Reagent Co., Ltd., Shanghai, China. Epichlorohydrin (ECH) was obtained from MACKLIN, Shanghai (China). Phosphate buffer saline (PBS) was purchased from Tianjin Huasheng Sci-Tech Co., Ltd., Tianjin (China). Yeast Extract and Peptone were purchased from Solarbio, Beijing (China). Sodium hydroxide (NaOH) was obtained from Damao Chemical Reagent Co., Ltd., Tianjin (China). The bagasse cellulose was provided by Guangxi Sugar Industrial Corp (Nanning, China). All products were used without further purification. Deionized H_2_O was used to prepare all solutions.

### 2.2. Synthesis of Cellulose Grafted with Polyhexamethylene Guanidine Hydrochloride

A method similar to the procedures reported previously was used [32]. Initially, equal molar amounts of hexanediamine (24.0 g) and guanidine hydrochloride (29.0 g) were mixed in a vacuumed three-necked flask under mechanical stirring (300 rpm) at 100 °C for 1 h, and then further reacted at 170 °C for 3 h. Ammonia, the byproduct of the reaction, was neutralized by dilute HCl solution. Afterwards, the viscous liquid was placed in a vacuum oven at room temperature to further remove residual ammonia. Finally, a solidified white polyhexamethylene guanidine hydrochloride (PHGH) was obtained. Antimicrobial modified cellulose was prepared by grafting the PHGH synthesized above bleached sugarcane bagasse pulp in the presence of epichlorohydrin as a coupling agent. Firstly, the epoxidation of cellulose was conducted according to our previous work [33]. Specifically, the sugarcane bagasse pulp was pretreated with NaClO_2_ and KOH to remove the lignin and hemicellulose from bagasse cellulose to obtain the sugarcane bagasse cellulose (SBC). 4 g of SBC was dispersed in 160 mL of diluted NaOH (2.5 wt %) solutions, using a high-speed homogenizer at 15,000 rpm. Next, epichlorohydrin (20 mL) and ethanol (40 mL) were dropwise added for epoxidation. Under mechanical stirring (400 rpm) at 60 °C, the reaction was maintained for 3 h. The product was washed and filtrate until pH = 7. The epoxidized cellulose (4.0 g) and deionized H_2_O (80 g) were mixed in a three-necked flask, and PHGH (2 g) was added to the epoxidized cellulose suspension above. The reaction lasted for 4 h at 70 °C and the resulting product was denoted as P-SBC. 

### 2.3. Hydrogel Preparation

To produce the antimicrobial hydrogels, 4 g of SBC cellulose and 96 g of 7 wt % NaOH/12 wt % urea/81 wt % water mixture were stirred extensively at −12 °C until the transparent solution (i.e., SBC solution) was formed. P-SBC was dissolved under the same conditions and mixed with SBC solution at different mass ratios. The hydrogel without antimicrobial modified-cellulose was denoted as S-CH, which is 10:0 dressing. After dissolution, ECH (as a crosslinker) was added into mixtures at 25 °C for 2 h to produce a homogeneous solution, which was then kept at 60 °C for 12 h to obtain hydrogels. Depending on the wt ratio of P-SBC to SBC, the hydrogel dressings were denoted as P-CH, 8:2 dressing or 6:4 dressing. The hydrogels without SBC were fabricated denoted as S-CH.

### 2.4. Polymer and Hydrogel Characterization

FTIR spectra were obtained with attenuated total reflectance (4000–400 cm^−1^ using 32 scans, Bruker TENSOR II, Karlsruhe, Germany) including background subtraction. 

The morphologies of cellulose and corresponding hydrogel were revealed using a scanning electron microscope under 5 kV accelerating voltage (Hitachi SU-8220, Tokyo, Japan). The hydrogel samples were first swollen in the deionized water to equilibrium, and then they were quickly frozen in liquid nitrogen thereafter. Next, these samples were further freeze-dried in a vacuum freeze drier until all the solvent was sublimed. Finally, the internal structure of hydrogel samples coated with gold for 40 s under a vacuum.

### 2.5. Compression Test

For compression test of hydrogels, the cylindrical samples were fabricated with a diameter of 25 mm and a length of 40 cm and tested using a universal testing machine (model 3367, Instron, Boston, MA, USA) at room temperature. The compression speed was 1 mm/s. Each type of the hydrogel sample was tested three times.

### 2.6. Swelling Characterization

The swelling properties of hydrogels were investigated in phosphate buffer saline. The freeze-dried hydrogels were cut into suitable size and weighted. The hydrogels were immersed in beaker with certain quantity of phosphate-buffered saline (0.05 mmol, pH= 7.4) at 37 °C to simulate the condition of the body. At designated time intervals, the hydrogel was removed from solution, gently wiped with filter paper to remove excess solution on the sample surface, carefully weighed, and immersed in solution again. The swelling process was repeated under the same conditions until the weight of hydrogels reached constant. Three specimens of each hydrogel were tested. The swelling degree (SD) of the hydrogels was estimated using the equation reported elsewhere [34].

$$SD(\%)=\frac{W_{t}-W_{d}}{W_{d}}\times 100$$
where *W_d_* is the weight of freeze-dried hydrogel, and *W_t_* is weight of swollen hydrogel at different time intervals.

### 2.7. Testing of Antimicrobial Activity

Employing a shaking flask method to evaluate the bactericidal effect of hydrogel samples, the antibacterial activities of samples against Gram-negative bacterium *Escherichia coli* (ATCC 8739) were determined. The hydrogels with and without P-SBC were prepared as previously described. The procedure was as follows: 10 mL of bacterial culture (10^5^ CFU/mL) was mixed with 0.2 g of chopped hydrogel and kept at room temperature until swelling of hydrogel in water reached equilibrium. Then, the mixtures were shaken at 200 rpm at 37 °C for 24 h. After shaking, different dilutions (e.g., 10^−1^, 10^−2^, 10^−3^, 10^−4^) were made by adding 1 mL of bacterial culture into 9 mL of PBS solution. Then, 0.1 mL of this culture was spread on an agar plate which was put into an incubator at 37 °C for 24 h. The number of colonies were counted, and three repetitions were conducted to obtain the average for each sample. The inhibition of the cell growth or antimicrobial rate was determined from the following expression:$$Growth\;inhibition\;of\;cell\;\left(\%\right)=\frac{A-B}{A}\times 100\%$$
where *A* = the number of the colonies counted from the control, *B* = the number of the colonies counted from the treated samples.

The qualitative test was assessed in a ring-diffusion method to evaluate the antimicrobial properties and to confirm the non-leaching effect of hydrogel dressing samples in this work. The procedures are as follows: 0.1 mL of bacterial culture (10^6^ CFU/mL) was spread on LB agar plates. The samples were carefully cut by punching to prepare round pieces of samples (about 10 mm in diameter) which were then plated on the surface of LB agar and incubated at 37 °C for 24 h. The observed inhibition zones can be used to assess the non-leaching characteristics of hydrogel.

### 2.8. Cell Viability

Cytotoxicity of hydrogels was evaluated using an MTT assay on NIH3T3 cells. After UV-sterilized samples for 60 min, the samples of P-CH hydrogels (4.0 mm × 4.0 mm) were immersed in phosphate buffer saline (0.01 M, pH = 7.4) at 37 °C until the swelling reached the equilibrium. NIH3T3 cells were seeded onto a 96 well plate at a density of 10^4^ cell/well and incubated at 37 °C with 5% CO_2_ atmosphere for 24 h. The culture medium and DMSO were used as blank and negative control, respectively. Then the cells of each well were treated with MTT (0.5 mg/mL) and incubated in an oven at 37 °C and 5% CO_2_ for 3 h, and then the 100 μL of culture medium was replaced with 100 μL of DMSO for each well. The cell viability treated with hydrogels was estimated in terms of percentage of cell viability. Microplate reader (Gen5, BioTeck, Seoul, Korea) was used to obtain the optical density (OD), where the absorbance was measured at λ = 550 nm after shaking the plate for 10 min. All assays were conducted in triplicate. 

## 3. Results and Discussion

### 3.1. Hydrogel Characterizations

#### 3.1.1. Antimicrobial Modification of Cellulose with PHGH

In this work, guanidine prepolymer was firstly synthesized by condensation polymerization [32]. Then, the prepolymer was grafted onto the epoxidized cellulose in the presence of ECH to produce antimicrobial-modified cellulose. The hydroxyl groups on cellulose main chains become oxygen anions under alkaline condition, which act as the nucleophilic reagent to react with epoxy group of epichlorohydrin. Specifically, the nucleophilic reagent firstly attacks C–O–C bond that is a position with a lower void steric resistance and a lower degree of substitution on the –CH_2_. The reaction follows the SN2 mechanism, owing to the epoxy group with high activity. Afterwards, there are two ways for crosslinking. The first one referring the guanidine-based polymer is due to the interaction between C=N and the imino groups. The π electron of C=N and the adjoining lone pair electron of imino groups form the electron hyper conjugation cloud and transform in the direction of leaving the lone nitrogen atom, making the hydrogen atom of nitrogen atom become active. The second one is attributed to the reaction of the imino groups far from the C=N in alkaline medium, which are easy to react with epoxy groups in the P-SBC. The schematic of the reaction is depicted in Scheme 1.

#### 3.1.2. Chemical Interactions of Functional Groups in Hydrogel

The crosslinking between SBC and P-SBC was further conducted using ECH as a crosslinking agent to manipulate the performance of the hydrogel as dressing in terms of antimicrobial activity and compression strength, etc. Because of the relatively low solubility of P-SBC in the solvent, the degree of crosslinking was not sufficiently high to form a stable hydrogel when P-SBC was used alone. The use of pure SBC as support is essential in the current work for forming the composite hydrogel via co-crosslinking. The schematic of the crosslinking reaction is depicted in Scheme 1.

The epoxy group of epichlorohydrin reacts with the hydroxyl group of cellulose to form an ether bond, while the other end of epichlorohydrin removes H–Cl to form a new epoxy group. With the continuation of crosslinking, the excess crosslinking agent was eventually converted to glycerol. Due to the limited solubility of the modified cellulose in sodium hydroxide and urea, less hydroxyl groups are exposed for the reaction, so that the degree of crosslinking is relatively low.

#### 3.1.3. FTIR of Cellulose and Hydrogel

The FTIR spectra of SBC and P-CH are shown in Figure 1. The peaks at 3390 cm^−1^ in the spectra of SBC are ascribed to the O–H stretching vibrations and the peak at 2899 cm^−1^ is the stretching vibration of aliphatic C–H. The peaks at 1159 and 1064 cm^−1^ are related to the C–O stretching vibrations of saccharide structure. The widened peaks in the region 3500–3000 cm^−1^ might be caused by stretching vibration of primary amine (N–H) and weak absorption peak of secondary amine in P-SBC and SBC. The peak of SBC at 1640 cm^−1^ belongs to the aromatic skeletal vibrations. The broad peak of P-SBC and P-CH at 1640 cm^−1^ is related to C=N deformation vibration [28,35]. All above changes indicated that SBC was successfully modified by PHGH.

#### 3.1.4. SEM Analysis

The morphologies of the cellulose and lyophilized hydrogels were observed by SEM (Figure 2), Figure 2a–c show a slight contrast in the surface morphology of the cellulose fibers before and after modified with PHGH. Cracks and burrs appeared on the smooth surface and the contact area became larger. The surface of cellulose changed due to grafting treatment.

From Figure 2d, the connected network structure in hydrogel can be clearly seen. All the hydrogel samples exhibited three-dimensional network structures, whereas sample S-CH appeared to be highly porous with 3D structures. The pore size of pure hydrogel is relatively uniform, and the pore diameter ranges between 10–30 μm. With the addition of P-SBC, the poor solubility of P-SBC in solvent lowered the degree of the polymer crosslinking, leading to uneven the pore channels.

The SEM images for S-CH and P-CH are presented in Figure 2e,f, respectively. As can been seen, both hydrogels showed interconnected porous structure, which is crucial for tissue regeneration, and considered to be the basic requirement for cell anchoring, proliferation, and new tissue growth on the surface of hydrogels. In addition, the large interconnected pores of these hydrogels are expected to increase water absorption and promote the diffusion of nutrients, biomolecules, and cellular waste products.

### 3.2. Swelling Properties

The SD (approximately 2000%) of S-CH was reached at equilibrium after the immersion of the hydrogel in phosphate saline buffer (PBS) at pH 7.4 and 37 °C for 24 h (see Figure 3). Similarly, the SD of P-CH also reached approximately 2000% after 24 h. At the beginning, the water absorption of pure hydrogel with a high degree of crosslinking (i.e., 10:0 S-CH) was lower than that of 8:2 and 6:4 P-CH. Gradually, more water molecules enter the hydrogel through the pore channels, allowing the high absorption capacity towards water. Finally, the swelling degree becomes approximately the same. The interaction of hydrogen bonds between molecules further strengthens the mechanical properties of hydrogel, but meanwhile lowers the swelling rate. When the hydroxyl groups on cellulose chains are connected with guanidine salt, the N**^…^**H–O hydrogen bonding between the P-SBC and SBC is created, owing to the presence of primary and secondary amino groups in P-SBC. Therefore, the antimicrobial hydrogels possess high water absorption capacity, applicable as wound dressing, in wet and warm environment conducive to skin wound repair [11,33].

### 3.3. Mechanical Properties

The mechanical strength of hydrogels is one of the important properties affecting the practical application of materials. As shown in Figure 4, the swollen hydrogel samples exhibited a “J” type of stress-strain curve, suggesting that the hydrogel possessed the high compressive strength [34]. The results indicated that S-CH appeared to be rigid. Its compressive stress increased initially, and the fracture occurred at shift around 20 mm with the yield compression stress reaching 61.35 KPa. With the increasing of the P-SBC, the compressive stress was decreased to 31.39 KPa.

In the hydrogel network, the rigid chains of SBC act as support. Because of its limited dissolution in NaOH/Urea solvent, antimicrobial modified cellulose hydrogel cannot maintain its stable structure without SBC present in the network.

### 3.4. Antibacterial Activity of P-SBC and P-CH

Guanidine-based polymer has excellent antimicrobial activities against extensive strains and the minimum inhibitory concentration (MIC) of guanidine-based polymer (PHGH in particular) against *E.coli* is 8 ppm [36,37]. Table 1 shows the antimicrobial activities obtained from the shaking flask method. 

P-SBC containing 0.5 wt % guanidine-based polymer showed a limited capability to deactivate *E.coli* with antimicrobial rate about 88.5%. With increasing the content of PHGH to 1.5 wt %, the excellent antimicrobial activity was achieved (the rate > 99.99%). Compared with the P-SBC, the P-CH containing 1.0 wt % PHGH also demonstrated an enhanced antimicrobial activity and the rate reached 97.6%, Similarly, when the P-CH content of PHGH was 1.5 wt %, the antibacterial rate reached almost 100%. The antibacterial ingredients in hydrogels are critical for preventing the dressing from being attacked by bacteria. Such a high antimicrobial activity is attributed to the PHGH grafted onto the fiber. The PHGH is a cationic polyelectrolyte (mainly due to C=N^+^ group), can readily adsorb on the negative-charge surface of bacterial cells by electrostatic interaction. The antimicrobial mechanism is governed by the physical action after the adsorption without raising drug resistant issues. The PHGH can exchange its ions with the potassium and magnesium in the cell membrane, thus alternating the fluidity and permeability of the cell membrane. The surface structure of the bacterial cell membrane is degenerated and destroyed. The membrane damage caused the leakage of low molecular weight cytoplasmic components and led to the collapse of the cells, thus deactivating the bacteria eventually.

Furthermore, Figure 5 presents the antimicrobial performance using the ring diffusion test. For sample P-CH (PHGH at 1.5%), there were no obvious bacteriostatic rings around the samples, whereas the results from shaking flask method has proved that the antibacterial efficiency of the same sample above 99.99%. No antimicrobial rings observed confirm the non-leaching characteristic of the antimicrobial hydrogel, which potentially eliminates the side effect of antimicrobial components in wound dressing and ensures the long-term effectiveness of bacterial barrier of the dressing. However, the hydrogel itself can act as a drug carrier, enabling the accommodation of drugs for healing purpose via controlled release. Meanwhile, the antimicrobial groups in the hydrogel prevent the dressing from being contaminated by bacteria surrounded.

### 3.5. Cytotoxic Characterization of P-CH Dressing

The cytocompatibility of hydrogel was characterized using an in vitro MTT method, which further validated the application prospect of hydrogel in the field of biomedicine as wound dressing and skin repair. This trial has been specifically designed to assess the mitochondrial function and cell viability and widely accepted to evaluate the cytotoxicity of new biomaterials in accordance with international standards (ISO 10993-5:2009).

Figure 6 shows the cell viability of the P-CH with the different contents of PHGH. The cell viability of S-CH was approximately 95%, and those of P-CH hydrogels containing 0.5, 1.0, 1.5, and 2.5 wt % of P-CH were approximately, 90.2%, 81.6%, 79.3%, and 75.7%, respectively. The results indicated that P-CH showed almost no cytotoxicity and good biocompatibility.

Clearly, the surface of the control sample (Figure 7a) was covered by cell almost completely, and the cells remained full, vigorous, regular shape, good adherent, uniform size, and strong diopter after the incubation for 24 h over direct contact. After direct contact with P-CH (Figure 7b), no significant difference was observed. Only a very small number of cells became irregular in shape and collapsed, and the cell fragments were slightly increased compared with the reference cell culture, which was consistent with the results of cell viability obtained from MTT assay. Therefore, the results suggested that the antimicrobial hydrogel is promising as green-based wound dressing due to its proper porous structure, high water retention, antimicrobial activity, and very limited cytotoxicity.

## 4. Conclusions

Novel biocompatible hydrogels as wound dressing were successfully prepared based on PHGH-modified cellulose using epichlorohydrin as a crosslinking agent. The resulting hydrogels with degree of swelling up to 2000% exhibited durable mechanical strength and high antimicrobial activity against *E. coli*. In the presence of 1.5% (wt) grafted PHGH in hydrogel, the growth inhibition towards *E.coli* reached over 99.99%. Moreover, the antimicrobial polymer was covalently bonded with cellulose backbones, which eliminated the harmful impact on the environment because of the non-leaching effects. This also ensures the durability of the antimicrobial effect. These three-dimensional hydrogels are biocompatible, leading to over 76% of the live cell viability responses of cells. Overall, the P-CH dressing based on bagasse cellulose hydrogel was well designed with the properties that are tunable through chemical modification and crosslinking. They are envisioned as promising scaffolds for use in chronic wound dressings and skin tissue repair applications.

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
