# Peer review of "Green-Based Antimicrobial Hydrogels Prepared from Bagasse Cellulose as 3D-Scaffolds for Wound Dressing"

_polymers, 2019, doi:10.3390/polym11111846_

Round 1

Reviewer 1 Report

This manuscript describes the fabrication of green based anitimicrobial hydrogel based on cellulose modified with guanidine polymer.The modification of cellulose structure with guanidine and fabrication procedure of antimicrobial hydrogel were well explained and their antimicrobial activity looks working. Thus, I recommend the publication of this manuscript after addressing the following issues.

    1. Overall, the quality of scheme and figures must be improved. For example, scheme 1 and 2 should be merged to clearly explain the fabrication of hydrogel based on chemically modified cellulose. Also, SEM images in Figure 2 don't explain well the explanation of hydrogel surface and structure.

    2. I recommend to show the picture images of nonleaching effect of composite hydrogels as a function of PHGH contents.

    3. Alive/dead staining experiments should be conducted to clearly demonstrate the biocompativility of composite    hydrogel. Furthermore, it might be better to demonstrate the heamocompativility of the composite hydrogel.

Author Response

Point 1: The quality of scheme and figures must be improved. For example, scheme 1 and 2 should be merged to clearly explain the fabrication of hydrogel based on chemically modified cellulose. Also, SEM images in Figure 2 don't explain well the explanation of hydrogel surface and structure. 

Response 1: We appreciate your valuable comments on the scientific merit of our manuscript.

We have merged the Schemes 1 and 2 in the revised manuscript.

Meanwhile, the sentences in section “SEM images in Figure 2” have been rewritten as follws: “From Figure 2d, the connected network structure in hydrogel can be clearly seen. All the hydrogel samples exhibited three-dimensional network structures; whereas sample S-CH appeared to be highly porous with 3D structures. The pore diameter of pure hydrogel is relatively uniform. With the addition of P-SBC, the poor solubility of P-SBC in solvent lowered the degree of polymer crosslinking, leading to the uneven pore channels.”

Point 2: I recommend to show the picture images of nonleaching effect of composite hydrogels as a function of PHGH contents.

Response 2: Duo to the fact that the contents of PHGH in P-CH were only set at two dosing levels, which is not sufficient to illustrate the nonleaching effect as a function of PHGH contents in the current work. However, this is an excellent suggestion, which will be taken into account in our future work.

Point 3: Alive/dead staining experiments should be conducted to clearly demonstrate the biocompativility of composite hydrogel. Furthermore, it might be better to demonstrate the heamocompativility of the composite hydrogel.

Response 3: Alive/dead staining experiments and the observation using an inverted phase contrast microscope were the common tests to visualize the status of cells. Based on the availability of laboratory facilities, we chose the inverted phase contrast microscope to observe and verify the change of tested cells. The test suggested by the reviewer will be considered in our future work.

Reviewer 2 Report

Dear Editor, dear Authors, Yuanfeng Pan et al. submitted a paper on the green synthesis of an antimicrobial hydrogel dressing. The authors modified sugarcane bagasse cellulose with guanidine-based polymer by in situ-grafting, using epichlorohydrin as a coupling agent, followed by crosslinking antimicrobial-modified cellulose with unmodified cellulose at various ratios. Afterward, they have characterized the resulting hydrogels by swelling measurements, compression test, Fourier transform infrared spectroscopy (FTIR) and scanning electron microscopy (SEM). The authors’ results demonstrated that the dressing possessed a degree of good swelling ~ 2000%, and a high compress strength ~31.39Kpa at 8:2 ratio of pristine cellulose to modified cellulose. Furthermore, they have assessed the hydrogels antibacterial activities of the dressing against E. coli using both shaking flask and ring diffusion methods; and their results demonstrated that the dressings were highly effective in deactivating bacterium without leaching effect. Moreover, they have assessed their hydrogels biocompatiblity with live cells, and found a viability of (NIH3T3) cells above 76%. For my opinion, the authors have performed an original research work that falls within the scope of Polymers MDPI Journal. However, for my opinion the manuscript is not well written, English style should be revised probably by a native English speaker, and results could be discussed further in a clearer manner. Therefore, I believe that the manuscript should be revised extensively before it can be accepted for publication in Polymers MDPI Journal.

I have the following comments to the authors represented below.

Comments:

Page 3, line 119 “pretreated sugarcane pulp (SBC)” authors should clarify what is SBC at this stage. Page 3, line 122 “The product was washed and filtrate until pH=7.” I cannot understand what the authors mean by this sentence please rephrase and clarify. Page 3, lines 123-124 “A certain amount of PHGH was added to the epoxidized cellulose suspension, followed by the reaction at the desired temperature and duration. The resulting product was denoted as P-SBC.” Please specify what was the amount used as well as the temperature, and for what duration Page 3, line 135 “4000–400 cm-1…” please correct -1 to superscript

Page 4, line 146 “The each type of hydrogel sample was…” please rephrase to read each type of the hydrogel sample was… Page 4, line 149 “PBS (phosphate saline buffer).” Use phosphate buffer saline Page 4, line 150 please correct weighted to read weighed also in line 153 Page 4 “Repeat the above process until the weight of hydrogels reached constant.” Please revise this sentence; I suggest that it should read the above process was repeated until a fixed weight is obtained? Page 4, line 155 “Three specimens of each hydrogel was tested.” Replace was tested by were tested Page 4, line 179 “…and after incubation at 37 °C for 24 h.” please rephrase, I suggest that it should read followed by incubation at 37 °C for 24 h. Page 4, line 183 “After UV-sterilized samples for 60 min,…” please rephrase Page 5, lines 187-188 “Then the cell was treated with MTT (0.5 mg/mL) incubated at 37 °C for 3 h, and the culture medium was then replaced with 600 μl/well of DMSO, which was dissolves enzymes in cells.” this sentence is not clear please rewrite with better English. I believe that the grammar and English I the manuscript needs a major revision Page 5, line 208 “The reaction scheme is shown in Scheme 1.” Similarly in the legend page 6, line 210. Please rephrase, avoid using scheme twice in the sentence

Scheme 1, I suggest that the authors write down each chemical formula the abbreviation used in the text (SBC, P-SBC and ECH) as they did in Scheme 2. Page 6, lines 217-218 “The use of pure SBC as support is essential in the current work for forming the composite hydrogel via co-crosslinking.” Please explain further and clarify why the use of SBC as support is essential Page 6 lines 221-222, “The epoxy group of epichlorohydrin reacts with the hydroxyl group of cellulose to form an ether bond, while the other end of epichlorohydrin removes H-Cl to form a new epoxy group. With the continuation of crosslinking, the excess crosslinking agent was eventually converted to glycerol.” The discussion here about the crosslinking reaction is not represented in the scheme 2; I think the authors should add those reactions to the scheme 2 or another scheme in supporting information. Page 6, line 231 “…and weak adsorption of secondary amine in P-SBC” what the authors mean by weak adsorption here? Is this a typing mistake and this should read absorption ! Page 6, lines 232-233, “The peak of SBC at 1640 cm-1 belongs to the aromatic skeletal vibrations; the broad peak of P-SBC and P-CH at 1640 cm-1 is related to C=N deformation vibration” how can the authors differentiate between the two peaks if both appear at 1640 cm-1 ?? Please clarify. Figure 1, I suggest that the authors add the chemical structures of a) PHGH, b) SBC, c) P-SBC and d) P-CH together with their abbreviation. Page 7, line 242, “From Figure 2d, the connected network structure in hydrogel can be clearly seen.” Please mark the connections on the figure with arrows Page 7 lines 242-243 “The pore diameter of pure hydrogel is relatively uniform and the pore diameter is about 10-30 μm.” Please revise the English in the sentence and void using pore diameter twice in the same sentence. Furthermore, show the pore diameter on the figure Page 7, line 244 “…the poor solubility of P-SBC in solvent..” specify what is the solvent Page 8, line 256 “The SD of S-CH approximately 2000% at equilibrium was reached…” Please revise the English and rephrase to clarify

Page 8, line 257 add space before Analogously, I suggest to replace analogously by similarly. Figure 3, I suggest that the authors add an insert in the figure showing the first 500 minutes difference discussed in the text by the sentence “At the beginning, the water absorption….” Page 8, line 260 “6:4 P-Gradually,” what the authors mean by P-Gradually please clarify Page 8, line 261 “Finally, the swelling degree is approximately the same.” Instead of finally state the time when the swelling becomes approximately the same Page 8 line 263 What the authors mean by twines ? Page 8, line 273 “As shown in Figure 4, the swollen hydrogel” remove space between the and swollen Figure 4 present the compressive stress strain in the order 10:0, 8:2, 6:4 instead of 8:2, 10:0, 6:4 Page 10 lines 305-306 “For sample P-CH (PHGH at 1.5%), and there is no obvious bacteriostatic circle around,…” Please rephrase and clarify Figure 6, the trend in the figure 6 shows that the increase in the percentage of P-CH from 0.5% wt to 2.5% wt decreases the cell viability from 95 % to about 75 %. However, the authors discuss this graph by “results indicated that P-CH showed almost no cytotoxicity and good biocompatibility.” I suggest that the authors should discuss the graph in figure 6 further. Moreover, Did the authors tried to check the cytotoxicity of a higher percentage of P-CH (>2.5 % wt) ?            Sincerely yours,

Author Response

Point 1: Page 3, line 119 “pretreated sugarcane pulp (SBC)” authors should clarify what is SBC at this stage. 

Response 1: Specifically, the sugarcane bagasse pulp was pretreated with NaClO2 and KOH to remove the lignin and hemicellulose from bagasse cellulose to obtain the sugarcane bagasse cellulose (SBC). 4g of SBC was dispersed in 160ml of diluted NaOH (2.5 wt%) solutions, using a high-speed homogenizer at 15,000 rpm.

Point 2: line 122 “The product was washed and filtrate until pH=7.” I cannot understand what the authors mean by this sentence please rephrase and clarify.

Response 2: pH=7 is corresponding to the conditions after the reaction involving NaOH (2.5 wt%) solutions for post-treatment to remove the residual NaOH.

Point 3: Page 3, lines 123-124 “A certain amount of PHGH was added to the epoxidized cellulose suspension, followed by the reaction at the desired temperature and duration. The resulting product was denoted as P-SBC.” Please specify what was the amount used as well as the temperature, and for what duration.

Response 3: We have rewritten this sentence, the experiment has been detailed as follows: “The epoxidized cellulose (4.0g) and deionized H2O (80g) were mixed in a three-necked flask, and PHGH (2g) was added to the epoxidized cellulose suspension above.”

Point 4: Page 3, line 135 “4000–400 cm-1…” please correct -1 to superscript.

Response 4: Thanks for pointing out the error. It is our negligence for this mistake. We have revised as follows:

FTIR spectra were obtained with attenuated total reflectance (4000–400 cm-1 using 32 scans, Bruker TENSOR II) including background subtraction.

Point 5: Page 4, line 146 “The each type of hydrogel sample was…” please rephrase to read each type of the hydrogel sample was…

Response 5: The phrase of “The each type of hydrogel sample was tested three times.” has been rephrased to “Each type of the hydrogel sample was tested three times.”

Point 6: Page 4, line 149 “PBS (phosphate saline buffer).” Use phosphate buffer saline

Response 6: “PBS (phosphate saline buffer)” has been rephrased to “phosphate buffer saline (PBS)”.

Point 7: Page 4, line 150 please correct weighted to read weighed also in line 153 Page 4 “Repeat the above process until the weight of hydrogels reached constant.” Please revise this sentence; I suggest that it should read the above process was repeated until a fixed weight is obtained? Page 4, line 155 “Three specimens of each hydrogel was tested.” Replace was tested by were tested.

Response 7: Thanks for pointing out the grammar mistakes. The corresponding sentences have been revised as follows: “At designated time intervals, the hydrogel was removed from solution, gently wiped with filter paper to remove excess solution on the sample surface, carefully weighed and immersed in solution again. The swelling process was repeated under the same conditions until the weight of hydrogels reached constant. Three specimens of each hydrogel were tested.”

Point 8: Duo to the swelling degree of the each hydrogel sample was undefined in a predetermined time. The phrase of fixed weight do not express that accurately.

Page 4, line 179 “…and after incubation at 37 °C for 24 h.” please rephrase, I suggest that it should read followed by incubation at 37 °C for 24 h. Page 4, line 183 “After UV-sterilized samples for 60 min,…”please rephrase.

Response 8: we have rewritten the relevant section as follows: “The samples were carefully cut by punching to prepare round pieces of samples (about 10mm in diameter) which were then plated on the surface of LB agar and incubated at 37 °C for 24 h. The observed inhibition zones can be used to assess the non-leaching characteristics of hydrogel.

(2.8. Cell Viability) Cytotoxicity of hydrogels was evaluated using an MTT assay on NIH3T3 cells. After sterilized under UV for 60 min, the samples of P-CH hydrogels (4.0 × 4.0 mm) were immersed in phosphate buffered saline (PBS, 0.01 M, pH = 7.4) at 37 °C until the swelling at equilibrium.”

Point 9: Page 5, lines 187-188 “Then the cell was treated with MTT (0.5 mg/mL) incubated at 37 °C for 3 h, and the culture medium was then replaced with 600 μl/well of DMSO, which was dissolves enzymes in cells.” this sentence is not clear please rewrite with better English. I believe that the grammar and English I the manuscript needs a major revision.

Response 9: Thank you for your suggestion. We have revised as follows: “Then the cells of each well were treated with MTT (0.5 mg/mL) incubated in an oven at 37 °C and 5% CO2 for 3 h, and then 100 μL of culture medium was replaced with 100 μl of DMSO for each well. The cell viability treated with hydrogels was estimated in terms of percentage of cell viability. Microplate reader (Gen5, BioTeck, Seoul, Korea) was used to obtain the optical density (OD), where the absorbance was measured at λ=550 nm after the plate was shaken for 10 min. All assays were conducted in triplicate.”

Point 10: Page 5, line 208 “The reaction scheme is shown in Scheme 1.” Similarly in the legend page 6, line 210. Please rephrase, avoid using scheme twice in the sentence.

Response 10: The sentence has been rephrased to “The schematic of the reaction is depicted in Scheme 1.”

Point 11: Scheme 1, I suggest that the authors write down each chemical formula the abbreviation used in the text (SBC, P-SBC and ECH) as they did in Scheme 2.

Response 11: The chemical formula and the abbreviation in Scheme 1 have been added in our revised manuscript.

Point 12: Page 6, lines 217-218 “The use of pure SBC as support is essential in the current work for forming the composite hydrogel via co-crosslinking.” Please explain further and clarify why the use of SBC as support is essential.

Response 12: The explanation has been provided as follows: “Because of the relatively low solubility of P-SBC in the solvent, the degree of crosslinking was not sufficiently high to form a stable hydrogel when P-SBC was used alone. The use of pure SBC as support is essential in the current work for forming the composite hydrogel via co-crosslinking. The schematic of the crosslinking reaction is depicted in Scheme 1.”

Point 13: Page 6 lines 221-222, “The epoxy group of epichlorohydrin reacts with the hydroxyl group of cellulose to form an ether bond, while the other end of epichlorohydrin removes H-Cl to form a new epoxy group. With the continuation of crosslinking, the excess crosslinking agent was eventually converted to glycerol.” The discussion here about the crosslinking reaction is not represented in the scheme 2; I think the authors should add those reactions to the scheme 2 or another scheme in supporting information.

Response 13: It is a well-known Williamson reaction which has been well documented.

Point 14: Page 6, line 231 “…and weak adsorption of secondary amine in P-SBC” what the authors mean by weak adsorption here? Is this a typing mistake and this should read absorption !

Response 14: Thanks for pointing out the typo. Indeed, “absorption” should be used herein instead of “adsorption” when the peak intensity in FT-IR is referred. The sentence has been rewritten to “the weak absorption peak of secondary amine in P-SBC”. The absorption intensity is relatively weak, compared with the one caused by the stretching vibration of primary amine (N-H).

Point 15: Page 6, lines 232-233, “The peak of SBC at 1640 cm-1 belongs to the aromatic skeletal vibrations; the broad peak of P-SBC and P-CH at 1640 cm-1 is related to C=N deformation vibration” how can the authors differentiate between the two peaks if both appear at 1640 cm-1 ?? Please clarify.

Response 15: From our previous work and those published by others (see the references 33 and 35 below), introducing PHGH into cellulose creates the characteristic absorption peak at 1640 cm-1 due to imino groups; and he peak of SBC at 1640 cm-1 belongs to the aromatic skeletal vibrations. Indeed, it cannot be differentiated between two compounds; however, the peak of P-SBC and P-CH at 1640 cm-1 are broaden, which implies the overlapping of the absorption peak or the co-existence of PHGH and SBC in the current system.

Wang, F.; Pan, Y.; Cai, P.; Guo, T.; Xiao, H. Single and binary adsorption of heavy metal ions from aqueous solutions using sugarcane cellulose-based adsorbent. Bioresour. Technol. 2017, 241, 482-490. Wei, D.; Li, Z.; Wang, H.; Liu, J.; Xiao, H.; Zheng, A.; Guan, Y. Antimicrobial paper obtained by dip-coating with modified guanidine-based particle aqueous dispersion. Cellulose 2017, 24, 3901–3910.

Point 16: Figure 1, I suggest that the authors add the chemical structures of a) PHGH, b) SBC, c) P-SBC and d) P-CH together with their abbreviation.Page 7, line 242, “From Figure 2d, the connected network structure in hydrogel can be clearly seen.” Please mark the connections on the figure with arrows.

Response 16: We have added the abbreviation in Figure 1, and marked the connections on the figure 2d with arrows. Please see the revised manuscript for details.

Point 17: Page 7 lines 242-243 “The pore diameter of pure hydrogel is relatively uniform and the pore diameter is about 10-30 μm.” Please revise the English in the sentence and void using pore diameter twice in the same sentence. Furthermore, show the pore diameter on the figure.

Response 17: We have shown the pore diameter on the figure in the revised manuscript. In order to avoid duplication in the same sentence, the phrase of “pore diameter” has been rephrased to “pore size”.

Point 18: Page 7, line 244 “…the poor solubility of P-SBC in solvent..” specify what is the solvent.

Response 18: The solvent was addressed in section 2.3 (Hydrogel Preparation). Specifically, 96g of 7wt%NaOH/12wt% urea/81wt% water mixture was stirred extensively at -12 °C until the transparent solution was obtained.

Point 19: Page 8, line 256 “The SD of S-CH approximately 2000% at equilibrium was reached…” Please revise the English and rephrase to clarify Page 8, line 257 add space before Analogously, I suggest to replace analogously by similarly. Page 8, line 260 “6:4 P-Gradually,” what the authors mean by P-Gradually please clarify Page 8, line 261 “Finally, the swelling degree is approximately the same.” Instead of finally state the time when the swelling becomes approximately the same.

Response 19: Thank you for suggestion. We have rewritten the relevant sentences as follows:

“The SD (approximately 2000%) of S-CH was reached at equilibrium after the immersion of the hydrogel in phosphate saline buffer (PBS) at pH 7.4 and 37 °C for 24 h (see Figure 3). Similarly, the SD of P-CH also reached approximately 2000% after 24 h. At the beginning, the water absorption of pure hydrogel with a high degree of crosslinking (i.e., 10:0 S-CH) was lower than that of 8:2 and 6:4 P-CH. Gradually, more water molecules enter the hydrogel through the pore channels, allowing the high absorption capacity towards water. Finally, the swelling degree becomes approximately the same.”

Point 20: Figure 3, I suggest that the authors add an insert in the figure showing the first 500 minutes difference discussed in the text by the sentence “At the beginning, the water absorption….”

Response 20: The insert suggested has been added in Fig. 3.

Point 21: Page 8 line 263 What the authors mean by twines ?

Response 21: It means that there are many hydrogen bonds associated with the S-CH. To avoid confusing, the wordings have been deleted. The next sentence has also been revised slightly. Please see the revised manuscript for details.

Point 22: Page 8, line 273 “As shown in Figure 4, the swollen hydrogel” remove space between the and swollen Figure 4 present the compressive stress strain in the order 10:0, 8:2, 6:4 instead of 8:2, 10:0, 6:4.

Response 22: It is our negligence for this mistake. we have revised the sentences as follows: “The mechanical strength of hydrogels is one of the important properties affecting their practical application. As shown in Figure 4, the swollen hydrogel samples exhibited a “J” type stress-strain curve, suggesting that the hydrogel possessed the high compressive strength [34].”

Point 23: Figure 4 present the compressive stress strain in the order 10:0, 8:2, 6:4 instead of 8:2, 10:0, 6:4

Response 23: Thanks for pointing out the error. “…in the order 10:0, 8:2, 6” has been described in the revised manuscript.

Point 24: Page 10 lines 305-306 “For sample P-CH (PHGH at 1.5%), and there is no obvious bacteriostatic circle around,…” Please rephrase and clarify.

Response 24: Thank you for your suggestion. The sentence has been revised as follows: “For sample P-CH (PHGH at 1.5%), there were no obvious bacteriostatic rings around the samples; whereas the results from the shaking flask method already proved that the antibacterial efficiency of the same sample was above 99.99%.”

Point 25: Figure 6, the trend in the figure 6 shows that the increase in the percentage of P-CH from 0.5% wt to 2.5% wt decreases the cell viability from 95 % to about 75 %. However, the authors discuss this graph by “results indicated that P-CH showed almost no cytotoxicity and good biocompatibility.” I suggest that the authors should discuss the graph in figure 6 further. Moreover, Did the authors tried to check the cytotoxicity of a higher percentage of P-CH (>2.5 % wt) ?

Response 25: The cytotoxicity test was conducted to further validate the potential of these hydrogels as wound dressings for various biomedical applications including skin repair. This test was specifically used to evaluate mitochondrial function and cell viability; and the testing method has been widely accepted to access the cytotoxicity of novel biomaterials according to the International Standard (ISO 10993-5:2009, Biological evaluation of medical devices: Tests for in vitro cytotoxicity). The data of the cell viability were all above 75%, demonstrating almost no cytotoxicity and good biocompatibility. When the P-CH contained 1.5wt% of PHGH, the wound dressing already showed excellent antimicrobial activity with the antibacterial rate at almost 100%. Therefore, the sample with a higher percentage of PHGH (>2.5 % wt) was not explored.

Round 2

Reviewer 1 Report

I support the publication of this manuscript in Polymers journal

Reviewer 2 Report

Dear Editor, dear Authors, Yuanfeng Pan et al. submitted a revised paper on the green synthesis of an antimicrobial hydrogel dressing. The authors have answered all the reviewer comments. For my opinion, the manuscript read much better and can be accepted for publication in Polymers MDPI Journal.

Comments:

Page 4, line 152 “The freeze-dried hydrogels were cut into suitable size and weighted.” please correct weighted to read weighed